# Adults with Cerebral Palsy: Navigating the Complexities of Aging

**DOI:** 10.3390/brainsci13091296

**Published:** 2023-09-08

**Authors:** Devina S. Kumar, Gabriel Perez, Kathleen M. Friel

**Affiliations:** 1Burke Neurological Institute, White Plains, NY 10605, USA; dek4004@med.cornell.edu (D.S.K.); gp5424@gmail.com (G.P.); 2Feil Family Brain & Mind Research Institute, Weill Cornell Medicine, New York, NY 10065, USA

**Keywords:** neurorehabilitation, barriers, quality of life, transitional care

## Abstract

The goal of this narrative review is to highlight the healthcare challenges faced by adults with cerebral palsy, including the management of long-term motor deficits, difficulty finding clinicians with expertise in these long-term impairments, and scarcity of rehabilitation options. Additionally, this narrative review seeks to examine potential methods for maintaining functional independence, promoting social integration, and community participation. Although the brain lesion that causes the movement disorder is non-progressive, the neurodevelopmental disorder worsens from secondary complications of existing sensory, motor, and cognitive impairments. Therefore, maintaining the continuum of care across one’s lifespan is of utmost importance. Advancements in healthcare services over the past decade have resulted in lower mortality rates and increased the average life expectancy of people with cerebral palsy. However, once they transition from adolescence to adulthood, limited federal and community resources, and health care professionals’ lack of expertise present significant obstacles to achieving quality healthcare and long-term benefits. This paper highlights the common impairments seen in adults with cerebral palsy. Additionally, it underscores the critical role of long-term healthcare and management to prevent functional decline and enhance quality of life across physical, cognitive, and social domains.

## 1. Introduction

Limited public and federal resources have increased the unmet needs of adequate health services, family education and clinical practice guidelines for adults with cerebral palsy. Cerebral palsy (CP) is an umbrella term used to describe motor and postural impairments caused by injury to the fetal or infant brain. The permanent brain lesion often coexists with epilepsy, as well as cognitive, sensory, behavioral, and secondary musculoskeletal disorders [1]. Additional surveillance questions developed by Surveillance of CP in Europe (SCPE) can be used to determine the inclusion and exclusion criteria for using CP as an umbrella term [2]. Although non-progressive, the accompanying impairments typically worsen with age. Thus, accurate diagnosis and early medical management from infancy is critical to develop motor milestones and maximize functional outcomes. Adults with CP have a similar life expectancy to the general population, but their economic impact is significantly higher. The estimated lifetime cost in the US is USD 15 billion with an average cost of 1.2 million per person with CP [3]. A review paper in 2023 identified 53 chronic conditions of which many were modifiable and preventative risk factors among adults with CP [4]. Although advancements in health care have improved treatment approaches, adults with CP face complex health decisions that impact their physical, mental, and participatory quality of life [5]. Thus, there is an urgent need to identify the optimal dosage of physical activity and exercise, and address the unmet needs for home, society, and community based services across the patients’ lifespans [6,7,8,9]. This narrative review will discuss the current health care needs, available interventions, and identify areas that require significant attention to address the quality of care in adults with CP. In this paper, we focus on ambulatory adults (GMFCS I-III) with CP, although some issues also apply to non-ambulatory adults. 

## 2. Causes of Cerebral Palsy

In recent years, an increased focus on interventions that promote perinatal and maternal well-being have been found with a lower incidence of CP [10,11]. The main known risk factors are birth weight deficiency and preterm labor [12,13]. Although many risk factors for CP have been identified, many are still unknown [14]. Risk factors of CP are classified according to the potential time of injury. Preconceptual risk factors come from mothers with pre-existing conditions like irregular menses, bacterial or viral infections, malnutrition, high blood pressure, diabetes, and thyroid conditions [15,16]. Prenatal risk factors cause 75% of all cases, which includes birth defects, intrauterine viral infections, inflammations, vascular failure, and multiple gestations [17,18]. Perinatal risk factors that occur during delivery are from birth asphyxia, prolonged delivery, use of forceps, or from low birth weight, maternal fever, and infection [13,15]. Neonatal and infant risk factors that increase the incidence of brain injury include trauma, congenital encephalopathy, jaundice, prolonged ventilation, and seizures [19,20,21]. 

## 3. Types of Cerebral Palsy

There are several ways to classify CP, each reflecting a different aspect of the disability. 

### 3.1. Motor Function

The Gross Motor Function Classification System (GMFCS) is a validated and reliable method that describes self-initiated movements made by a person with CP [22,23]. It classifies individuals into one of five categories. 

GMFCS I: the person is ambulant and can walk indoors and outdoors, run, or jump with small limitations in gross motor function. 

GMFCS II: the person is ambulant but requires a railing, handheld mobility device, or assistance while climbing stairs, walking long distances, or walking on uneven surfaces. 

GMFCS III: the person requires a wheeled mobility device for daily activities or needs a manual wheelchair for long distances. 

GMFCS IV: the person typically uses wheeled mobility in most environments and may use body support walkers to mobilize in some environments.

GMFCS V: the person is completely dependent on assistance in daily activities. Some might achieve self-mobility using a powered wheelchair with extensive adaptations [23]. 

### 3.2. Muscle Tone

The SCPE classification system was developed to categorize CP based on muscle tone (high, variable, or low), dyskinetic tone (dystonia or choreo-athetotic), ataxia, and non-classifiable [24]. Approximately, 82% of the CP population has spasticity caused by injury to the pyramidal tract [14]. Approximately, 15% have extrapyramidal injuries that cause dyskinetic syndromes. The movements are described as slow, involuntary, or jerky in nature [25,26]. A small percentage of people with CP have cerebellar injuries that cause ataxia characterized by unsteady gait, limb incoordination, and difficulties with fine motor movements [27].

### 3.3. Regions of the Body Affected 

CP sub-types are classified by the topographic distribution of impairments in the body. In addition, 35% of the CP population has spastic diplegia where the lower limbs are more affected than the upper limbs [20], while 25% have hemiplegia affecting one side of the body and 20% have quadriplegia wherein all limbs are affected. A small percentage of the population with one limb involved will be diagnosed with monoplegia. 

These classifications are combined when giving a label to a type of CP. For example, if a child can run and jump, has increased tone, and has impairments restricted to one side of the body, the child would be classified as having GMFCS I spastic hemiplegia. 

## 4. Ageing with Cerebral Palsy

Improvements in health care have helped lengthen the lifespan of individuals with CP. Most people with CP have a typical life expectancy. Yet, without ongoing rehabilitation, physical, cognitive, and cognitive skills are not fully maintained across adulthood [28,29,30] [Figure 1]. Research and care for people with CP are targeted towards children to optimize their quality of life from early years. Approximately 50% of individuals with CP report a decrease in daily functional activities by the fourth decade of life [28]. The exponential decrease in rehabilitation opportunities during adulthood is caused by many factors. A recent study found that many people with CP are sedentary for more than 75% of their waking hours and spend less than 20% on light-intensity activities [31]. There is limited research to determine optimal exercise and therapeutic interventions for adults with CP. Physicians and individuals with CP often do not know strategies to maximize the health of adults with CP. As a result, many adults with CP are advised that rehabilitation beyond childhood is unavailable or ineffective. 

### 4.1. Musculoskeletal Complications

Adults with CP experience premature deterioration and deconditioning of the musculoskeletal system compared to people without CP. Approximately 25–30% of adults with CP in their 20s and 30s report deterioration in gross motor function compared to childhood [32,33]. Left untreated, this can cause hip dislocation associated with pain, scoliosis, and decreased mobility [34,35]. Hip stability is strongly correlated with the ability of a person with CP to walk [36]. Therefore, dietary, physical fitness, and medical strategies to improve bone mineral density and early hip surveillance are important to minimize neurodegenerative hip changes that could limit functional mobility [9,37,38,39]. 

Adults with more severe mobility restrictions have a highly increased risk of developing musculoskeletal morbidities compared to those without CP [40]. Contractures develop over time, increasing joint stress, muscle pain, and risk of falls [40,41,42]. Adults with CP have reported fear, humiliation, and decreased self-esteem after falls at home and in public places [42,43]. Subsequently, this can lead to an increased risk of settling into a sedentary lifestyle at home.

### 4.2. Cancer

Breast cancer mortality in women with CP is three times that of the general population [44]. Late mammography, limited community awareness, nulliparity, poor interactions with health professionals, and limited adaptive hospital equipment for evaluations prevent many people from seeking preventive care. Individuals with CP have greater mortality risks from all causes of cancer, except lung cancer, compared to the general population. Therefore, early detection can decrease the mortality rate of women by 40% within the age group of 50 to 69 [45]. Improving community awareness will enable the early detection and management of commonly undiscussed health risks like cancer [46]. 

### 4.3. Chronic Pain and Fatigue

Chronic pain and fatigue are the most common causes for deterioration in walking for adults with CP. Almost 80% of adults with CP experience severe pain and fatigue that worsen over time [40]. Back, shoulder, and hip pain are the most common, with approximately 50% of the population experiencing pain in more than two joints [41,47]. Several factors contribute to pain and fatigue, including the severity of CP, nutritional health, bone density, and asymmetric gait. These factors can also worsen motor function over time [42,48,49]. Furthermore, pain decreases the ability of individuals to maintain a healthy lifestyle [50]. 

Since pain and fatigue can have a profound impact on quality of life, pain management is essential for adults with CP. Isolated stretching, Tai Chi, and Pilates improve flexibility, range of motion and are associated with positive cognitive and behavioral outcomes [41,51,52]. Preventive strategies and medical management can decrease pain and modulate energy expenditure to reduce fatigue. 

### 4.4. Spasticity

Spasticity is characterized by increased muscle tone with sudden resistance to changes to joint movement speed during passive range of motion [53,54]. It leads to various secondary impairments including contractures, muscle stiffness, sustained muscle spasms, pressure ulcers, and increased muscle stress. Secondary musculoskeletal and neurological conditions can lead to premature ageing of adults with CP, which is characterized by an early decrease in mobility and function compared to the general ageing population [55]. Progressive chronic conditions in adulthood are related to increased spasticity and mobility impairments [48,49]. Therefore, adults with CP require spasticity management to reduce injuries of muscles and joints and minimize the accelerated decline in functional performance. 

Although spasticity management begins in infancy to prevent or minimize postural impairments, it requires life follow up. Rehabilitation therapy, oral (Lorazepam) and intra-thecal medications (Baclofen), botulinum toxin (Botox), and neurosurgical procedures like dorsal root rhizotomy are standard treatment methods [39]. However, all treatment approaches have limitations. Oral medicines have limited efficacy and side effects. Intra-thecal medicines often lose efficacy over time, requiring higher doses, and cause systemic side effects. Botox can cause irreparable damage by affecting the motor neurons or causing permanent hypotonia [56,57,58,59]. 

### 4.5. Mental Health

The focus on maintaining physical, social, and mental health function decreases with transition into adulthood. Depression and anxiety are more prevalent compared to the general population [60]. Adults with CP are also at a higher risk of developing dementia and Alzheimer’s disease [61]. Other mental health disorders that can affect daily lives include personality and behavioral, alcohol-related, and opioid-related disorders. Approximately 2.8% have schizophrenic disorders and 19.5% have mood affective disorders [62]. Therefore, mental health is not only linked to multiple complications but also considered a major burden to adults with CP. In addition, psychological well-being has a significant impact on depression, mobility, autonomy, sleep, and mood [51,63]. If left untreated, these cumulative health-related disorders can cause significant behavioral changes and disrupt current and future society and interpersonal relations [64,65]. 

Optimal quality of life in adults with CP requires an increased focus on comprehensive mental health. Individuals with intellectual disabilities may be more susceptible to developing serious mental health issues, in part due to the difficulty of individuals reporting symptoms and the lack of focus on mental health by caregivers and clinicians [66]. Early support and access to health services, as well as a framework for assessing challenging behaviors and possible treatment options, are needed [62,67]. Ultimately, awareness of the importance of physical and mental healthcare can provide a comprehensive approach for wellness and quality of life.

### 4.6. Women’s Health 

Women with disabilities face different challenges in reproductive health care from pre- to post-natal periods. Although women with and without disabilities share common fertility rates, only 11% of those with disabilities become pregnant [68]. This disparity suggests that there is a lack of health care services, family support, adequate accommodation, and experience of practitioners in pre- and post-pregnancy care for women with CP [69,70]. Common misconceptions by health care professionals on the sexual lifestyle of women decrease the incidence of experiencing motherhood [71,72]. Additionally, women with disabilities often experience situations with skilled health professionals who may discourage pregnancy due to perceptions that women with CP cannot complete a healthy pregnancy and become successful mothers [73,74,75]. 

Discrimination and negative perceptions of women with CP by health professions can be addressed through outreach interactive programs that include education about the importance of lifelong medical care [30,42,76]. This would strengthen relationships between people with CP and their providers. At administrative levels, hospital management and policy makers must be encouraged to improve structural changes in health centers, transport, and communication services across the disability spectrum.

## 5. Benefits of Rehabilitation 

Attention to health care and rehabilitation for adults with CP has increased in the last decade. However, many gaps remain in their health care continuum, from adolescence to late adulthood. Table 1 summarizes some key factors and barriers to healthy ageing from individual, environmental, and health care perspectives [8,28,29,77,78,79,80,81,82,83,84]. Chronic conditions are strongly associated with early ageing and significant mobility impairments in adults with CP [85]. Adults with CP have a greater risk of fractures independent of osteoporosis or cardiometabolic factors [86]. Arthritis, pain, and chronic fatigue establish a downward spiral of functional activities affecting quality of life. To prevent deterioration of physical, mental, and social domains of life, certain lifestyle modifications are crucial. 

Health professionals often have limited knowledge on the preventive strategies and medical management needed to optimize health care for adults with CP. Lack of experience treating adults with CP by clinicians often means that disease-specific risk factors and knowledge-based consultation go unnoticed [64,65]. Professional specialists can enhance health services by improving their knowledge on intervention practice to gain experience, discuss prevention, and consider rehabilitation strategies. By encouraging active dialogue with support groups and outreach programs, providers would not only understand the unmet needs of families but also use educational resources for better practice. 

Decreased bone density combined with falls are a significant cause of fractures in adults with CP [42,86,87]. Adults with CP are at a higher risk to develop osteoporosis and four times more likely to fall compared to adults without CP [87]. Therefore, early monitoring of risk factors and periodic scanning of bone health and bone mineral density using Dual Energy X-Ray Absorptiometry (DEXA) may assist with the early management of osteoporosis and arthritis [33,86]. Recommended interventions include fall prevention programs, muscle strengthening and nutritional supplements to enhance bone health and muscle strength in adults with CP. Specifically, balance training, Progressive Resistive Exercises (PRE), treadmill walking without support, and split-belt treadmill perturbations can improve static and dynamic stability and functional mobility [39,88,89,90,91]. 

Early investigation of cardiovascular and metabolic fitness levels is crucial for physical endurance, agility, muscle strength and muscle power. Individuals with CP are at a heightened risk of developing cardiometabolic conditions that accelerate immobility, multiple morbidities and decreased functional capacity [92,93]. 

Recently, cardiorespiratory (aerobic) exercise training guidelines specifically for the CP population have recommended exercising for at least 20 min, 3 times per week at >60% of the patient’s peak heart rate, or between 46 and 90% VO_2_ peak, and resistance exercise (anaerobic) training for at least 12–16 weeks, 2–4 times/week in sets of up to 3 (6–15 repetitions/set) of 50–85% maximum repetitions. PRE in children with CP is recognized to improve muscle strength and gait function [94,95,96]. In ambulant adults (GMFCS I–III), PRE improved muscle strength and its effects remained for at least 11 weeks compared to no intervention. However, the systematic review also indicated that muscle strength was not affected by PRE intensity or training volume [97]. Further investigation into exercises that also improve bone health is needed [9]. For those with severe mobility impairments, replacing sedentary activity with light exercise on daily basis can help reduce cardiovascular complications and improve general health [31]. Thus, strength programs tailored to an individual’s fitness and sedentary lifestyle may provide the best outcomes for adults with disabilities. 

### Combinatorial Intervention Approaches

Even the best available interventions for children or adults with CP do not produce the robust improvements in function or quality of life that people with CP deserve. It is likely that combinatorial interventions have the strongest promise to optimize function in a person with CP. Greater work is needed to decide which combinations are the most effective for which patients, and to expand upon the availability of effective combinatorial interventions. Among the combinatorial interventions, combined PRE and functional anaerobic training in young adults with CP significantly improved lower limb muscle volume associated with increased functional capacity and muscle strength; virtual reality with conventional physical therapy improves upper extremity function [98]; dance and rhythmic auditory movements enhances gait and balance [99], and rhythmic auditory stimulation with complex cords significantly improved joint angle and range of motion [100]. However, to strengthen the current research, it is likely that motor training will need to be paired with another therapeutic strategy, such as pharmacological or non-invasive brain stimulation. Only one case report has been published on adults with CP, combining NIBS with motor training on a robotic device [101]. While the findings were promising, much more work is needed to explore the use of different combined approaches as therapies for adults with CP.

## 6. Conclusions

While early intervention in children with CP is important for optimal development, proactive continuation of care through adulthood is essential. For adults with CP, unfortunately, a lack of medical and rehabilitative care typically leads to a range of musculoskeletal, respiratory, and related complications. Thus, there is an urgent need to improve medical and community services to optimize health and quality of care across these patients’ lifespans. Future development of affordable accessible fitness facilities, accessible health care services, and training programs for medical providers could support healthier lives of adults with CP. 

## Figures and Tables

**Figure 1 brainsci-13-01296-f001:**
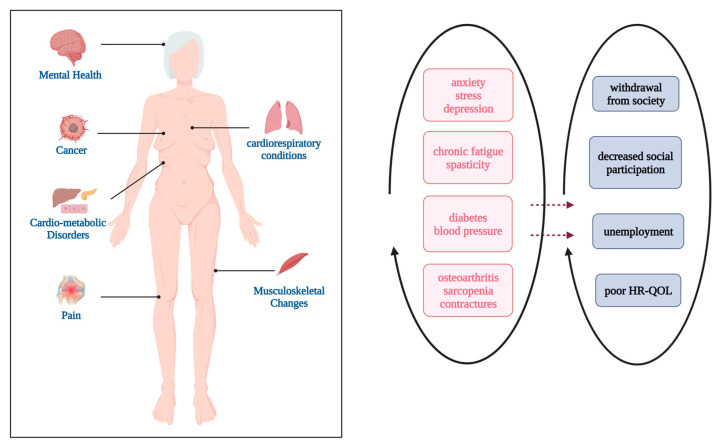
Non-neurological effects of ageing with cerebral palsy (created with BioRender.com).

**Table 1 brainsci-13-01296-t001:** Barriers to healthy aging in adults with cerebral palsy.

Sub-Domains	Key Factors	Barriers
Individual	Physical	lack of home-based adaptive exercise equipment bathroom modifications as permitted by the Americans with Disabilities Act
	Mental	low motivation and energy to exercise pain, anxiety, depression fear from falls/injury
	Social	limited inter-personal communication with neighbors social stigma lack of social support
	Economic	all therapy services lack of cost-effective equipment to assist with activities of daily living
Environmental	Physical	narrow residential entrances and hallways, stairs transportation and parking at community and fitness centers non-adaptive equipment for physical activity and exercise lack of curbs, parking, and poor infrastructure
	Mental	negative experience from seeking help limited mental health care coverage low self confidence among peers
	Social	lack of community awareness programs and opportunities stigma, prejudice, and discrimination
	Economic	limited state and federal funding to community centers lack of funds for community awareness and socialization
Health Care	Physical	limited access to healthcare centers lack of adaptive medical equipment inadequate accommodations during screening and diagnosis
	Mental	mistrust of medical professionals and services fear of not being understood
	Communication	negative attitude of health care service providers lack of awareness for treatment options limited clinical experience with treating adults with CP limited awareness of available resources to facilitate decision making
	Economic	limited funding mechanisms to support ongoing care inadequate policy development to enhance structural and professional reorganization

## Data Availability

No new data were created or analyzed in this study. Data sharing is not applicable to this article.

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
