# Peer review of "Adults with Cerebral Palsy: Navigating the Complexities of Aging"

_brainsci, 2023, doi:10.3390/brainsci13091296_

Round 1

Reviewer 1 Report

Comments are attached

A few misspellings and grammatical errors. A few run-on sentences.

Author Response

REVIEWER #1

General Concept Comments

A few pivotal “older “ articles are fine for setting the background, however, there are sufficient articles published within the last five years that should form the basis for the content in this review. A few examples:

  • Prevalence and incidence of chronic conditions among adults with cerebral palsy: A systematic review and meta-analysis. Ryan JM, Albairami F, Hamilton T, Cope N, Amirmudin NA, Manikandan M, Kilbride C, Stevenson VL, Livingstone E, Fortune J. Dev Med Child Neurol. 2023 Sep;65(9):1174-1189. doi: 10.1111/dmcn.15526. Epub 2023 Feb 20. PMID: 36807150 Review.
  • Risk of Depression and Anxiety in Adults With Cerebral Palsy. Smith KJ, Peterson MD, O'Connell NE, Victor C, Liverani S, Anokye N, Ryan JM. JAMA Neurol. 2019 Mar 1;76(3):294-300. doi: 10.1001/jamaneurol.2018.4147. PMID: 30592485
  • Designing exercise to improve bone heath among individuals with cerebral palsy. Gannotti ME, Liquori B, Thorpe DE, and Fuchs RK. Pediatric Physical Therapy: 2021; 33(1): 50-56.
  • Meeting the unmet needs of disabled adults with cerebral palsy. Werner S. Dev Med Child Neurol. 2022 Oct;64(10):1187-1188. doi: 10.1111/dmcn.15249. Epub 2022 Apr 21. PMID: 35451083
  • Health service use among adults with cerebral palsy: a mixed methods systematic review protocol. Manikandan M, Walsh A, Kerr C, Walsh M, M Ryan J. BMJ Open. 2020 Aug 30;10(8):e035892. doi: 10.1136/bmjopen-2019-035892. PMID: 32868352

RESPONSE: Thank you for the suggestion. The introduction section has been updated to include more recent publications including the above citations (lines 40-42, lines 43-46)

  1. Please check the manuscript for spelling and grammar errors.

RESPONSE: Thank you for your feedback. We have re-checked the manuscript for spelling and grammatical errors.

  1. Please use “people first” language throughout manuscript. For example: NOT “CP literature” BUT “literature on strength training that involves individuals with CP” (lines 224-225)

RESPONSE: Thank you for the suggestion. We have revised the full paragraph. The sentence on “CP literature..” has been removed.

Specific Comments

  1. Under the subtitle “Types of Cerebral Palsy” the paragraph is very “dense” and the content might be better understood by presenting the information in table format. In addition, it is recommended that authors also describe “functional” levels a defined by the GMFCS. This classification system is widely used and recognized in research involving individuals with CP. It will also be easier for a reader unfamiliar with the “types of CP” to understand functional ability.

RESPONSE: Thank you for the suggestion. We considered the feedback from all the reviewers and decided to present the data in a text format that is easily understandable (lines 66-104). For a figure representation, it would not have been possible to get permission from the original authors of the GMFCS paper within 8 days.

  1. Recommend using the word “typical” as opposed to “normal” throughout manuscript.

RESPONSE: Thank you for the suggestion. The word “normal” has been replaced with “typical” throughout the text.

  1. Under 4.2 Cancer: consider using “preventative care” to replace “medical Check-ups”.

RESPONSE: Thank you for the suggestion. The word “medical check-up's” has been replaced with “preventive care” (line 141).

  1. Consider moving section 4.3 Discrimination in health care down in manuscript under 4.7 Women’s Health. The topic of discrimination is very important and must be addressed and might carry a more powerful impact if it was the culminating content in Ageing with Cerebral Palsy section.

RESPONSE: Thank you for the suggestion. We have moved the paragraph on discrimination in health under womens health (lines 210-215)

  1. In section 4.5 Spasticity authors mention “premature ageing” for the first time in the manuscript. A definition and relevant references would provide reader a better understanding of the term.

RESPONSE: Thank you for the suggestion. We have provided a definition and cited a reference for premature ageing in CP (lines 164-167).

  1. Table 1: Under Individual/Physical/ Barriers: would “narrow residential entrances and hallways, stairs” better fit under Environmental/Physical/Barriers?

RESPONSE: Thank you for the suggestion. We have moved “narrow residential entrances and hallways, stairs” under “Environmental/physical/Barriers” in Table 1.

  1. Table 1: Under Individual/Economic/Barriers: Should “limited physical therapy sessions covered by insurance” be more inclusive if it included “all therapy services”?

RESPONSE: Thank you for the suggestion. We have replaced the above with “all therapy services” in Table 1.

  1. Lines 194-196: There is some recent literature on health services research for adults with CP that might compliment this content.

RESPONSE: Thank you for the suggestion. We have added citations to address the content on health service.

  1. Line 199 requires a relevant reference. Not sure bone structure has been found to be a significant predictor of falls in this population?

RESPONSE: Thank you for the suggestion. We have replaced “bone structure” with bone health and added context to the initial statement (lines 238-241).

  1. Lines 200-202: Ref # 59 does not seem to be a relevant reference for this content?

RESPONSE: Thank you for the suggestion. The reference has been removed.

  1. Line 205: “split-belt treadmill perturbations” would be the correct term to replace “moving floor perturbations”.

AND

  1. Lines 204-206: A reference is required. There is recent relevant literature on using split-belt treadmill training to help improve gait symmetry and prevent falls in adults with CP.

RESPONSE: Thank you for the suggestion. “Moving floor perturbations” has been replaced with “split-belt treadmill perturbations” and cited accordingly (line 246).

  1. Line 210: References 60 and 61 relate to content. However, references 62-63 are references related to intervention and exercise and are not appropriate references to be included with 60-61.

RESPONSE: Thank you for the suggestion. Reference 62 and 63 have been omitted.

  1. Lines 210-213: Require a reference. There is literature on aerobic and anaerobic fitness measures for adults with CP in literature.

AND

  1. Line 223: References need verified. Ref 65 does not relate to this content? Ref 66 does not exist in reference list? There are numerous refs related to progressive resistive exercises for adults with CP on the literature.

RESPONSE: Thank you for the suggestion. We had re-written the section and focused on aerobic, anerobic, and progressive resistive exercises with appropriate citations (lines 252-263).

  1. Section 5.1 Combinatorial intervention approaches: There have been several “combinatorial intervention approaches in cited in literature that should be included. A few are below.
  • Functional Anaerobic and Strength Training in Young Adults with Cerebral Palsy. Gillett JG, Lichtwark GA, Boyd RN, Barber LA. Med Sci Sports Exerc. 2018 Aug;50(8):1549-1557. doi: 10.1249/MSS.0000000000001614. PMID: 29557839. Clinical Trial
  • Effectiveness of virtual reality in children and young adults with cerebral palsy: a systematic review of randomized controlled trial. Fandim JV, Saragiotto BT, Porfírio GJM, Santana RF. Braz J Phys Ther. 2021 Jul-Aug;25(4):369-386. doi: 10.1016/j.bjpt.2020.11.003. Epub 2020 Dec 5. PMID: 33358737
  • Dance and rehabilitation in cerebral palsy: a systematic search and review. López-Ortiz C, Gaebler-Spira DJ, Mckeeman SN, Mcnish RN, Green D. Dev Med Child Neurol. 2019 Apr;61(4):393-398. doi: 10.1111/dmcn.14064. Epub 2018 Oct 23. PMID: 30350851
  • Gait training for adults with cerebral palsy following harmonic modification in rhythmic auditory stimulation. Kim SJ, Yoo GE, Shin YK, Cho SR. Ann N Y Acad Sci. 2020 Aug;1473(1):11-19. doi: 10.1111/nyas.14306. Epub 2020 Apr 30. PMID: 32356332

RESPONSE: Thank you for the suggestion. We have included the above references under section 5.1 (lines 272-278)

  1. Conclusions: A few awkward, run on sentences. Make sure you emphasize “adults” throughout this paragraph to ensure reader concludes that it is “adults with CP” that are getting disproportionate care, intervention etc. as they age.

RESPONSE: Thank you for the suggestion. We have revised the entire conclusion section and emphasized on adults with CP as the target population of this paper.

Reviewer 2 Report

This is an important topic - that has not been comprehensively addressed in the literature - and is relevant to the scope of this journal. The abstract is well written and I was looking forward to reading the results and discussion.

The article has been submitted under the category review - and described as a review - however this is misleading. As currently written, this manuscript falls somewhat between an opinion piece and a narrative review. No methodology for the review or search is provided - or even a detailed justification of why and how the topics included were identified, and that the references cited are the most important or up-to-date.  The manuscript jumps from a very brief introduction into some background on causes of CP and types of CP to ageing with CP.

This is disappointing - as the topics selected (pain, fatigue, reproductive health, discrimination in health care provision etc.) would be important additions to the literature. 

Why is spasticity a topic - but dystonia is not even mentioned?

Under the description of types of CP, the terms mild, moderate and severe are used (without justification or referencing). This is a value judgement and difficult to quantify. I would suggest using functional descriptors such as ambulant, ambulant with aids, non-ambulant - and linking this with GMFCS levels and descriptors- with references relevant to use in adolescence and adulthood.  

It also appears that most of the paper is considering the needs of individuals who are at least partly ambulant. If this is the case, then this should be made clear. If the intention is to address these topics for all GMFCS levels - then clarification of differences for different functional abilities is needed. For example, bone mineral density and impact on other body systems are more of an issue for individuals with non-ambulant CP. 

References: There are errors in the referencing that make it difficult to judge whether important references are missing. For example, the Spasticity section has reference numbers 36-42 - while in the reference list the references on this topic are 34-40; mental health should be 41-48; and Women’s Health should be 49-54 - Are there two references missing?

I anticipated a more comprehensive discussion of healthcare issues and use of healthcare/rehabilitation from different international perspectives.

The rehabilitation section also appears to be under-referenced.

For example, a review of exercise interventions for bone health for individuals with CP was published in 2021 by Gannotti.

Are there any surveys or qualitative studies that would provide data regarding difficulties of rehabilitation interventions or healthcare provision for adults with CP?

Table 1 illustrates a number of barriers - while this is useful - where is the data that supports this table?  Is this your opinion - or is it the synthesis of information from different studies?

I think this manuscript has potential to be very useful - but it needs major revision as a review.

Author Response

REVIEWER #2

This is an important topic - that has not been comprehensively addressed in the literature - and is relevant to the scope of this journal. The abstract is well written and I was looking forward to reading the results and discussion.

 The article has been submitted under the category review - and described as a review - however this is misleading. As currently written, this manuscript falls somewhat between an opinion piece and a narrative review. No methodology for the review or search is provided - or even a detailed justification of why and how the topics included were identified, and that the references cited are the most important or up-to-date.  The manuscript jumps from a very brief introduction into some background on causes of CP and types of CP to ageing with CP.

This is disappointing - as the topics selected (pain, fatigue, reproductive health, discrimination in health care provision etc.) would be important additions to the literature. 

Why is spasticity a topic - but dystonia is not even mentioned?

RESPONSE: Thank you for the suggestion. We have included “dystonia” under the SCPE classification (line 86).

Under the description of types of CP, the terms mild, moderate and severe are used (without justification or referencing). This is a value judgement and difficult to quantify. I would suggest using functional descriptors such as ambulant, ambulant with aids, non-ambulant - and linking this with GMFCS levels and descriptors- with references relevant to use in adolescence and adulthood.  

RESPONSE: Thank you for the suggestion. We have edited the section on types of CP to include the GMFCS, SCPE, and topographical classification (lines 66-104)

It also appears that most of the paper is considering the needs of individuals who are at least partly ambulant. If this is the case, then this should be made clear. If the intention is to address these topics for all GMFCS levels - then clarification of differences for different functional abilities is needed. For example, bone mineral density and impact on other body systems are more of an issue for individuals with non-ambulant CP. 

RESPONSE: Thank you for the suggestion. We have included a sentence in the introduction section which mentions the focus of the review is on ambulant adults (GMFCS I-III). Lines 48-49

References: There are errors in the referencing that make it difficult to judge whether important references are missing. For example, the Spasticity section has reference numbers 36-42 - while in the reference list the references on this topic are 34-40; mental health should be 41-48; and Women’s Health should be 49-54 - Are there two references missing?

RESPONSE: Thank you for the suggestion. We have gone through and re-checked the references to match the text.

I anticipated a more comprehensive discussion of healthcare issues and use of healthcare/rehabilitation from different international perspectives.

RESPONSE: Thank you for the suggestion. The care of adults with CP is highly variable around the world. Properly summarizing these differences would require its own manuscript.

The rehabilitation section also appears to be under-referenced.

RESPONSE: Thank you for the suggestion. We have added more text and references to support the rehabilitation section.

For example, a review of exercise interventions for bone health for individuals with CP was published in 2021 by Gannotti.

RESPONSE: Thank you for the suggestion. We have cited Ganotti’s work in the introduction section and under musculoskeletal complications.

Are there any surveys or qualitative studies that would provide data regarding difficulties of rehabilitation interventions or healthcare provision for adults with CP?

RESPONSE: Thank you for the suggestion. Yes, we have included references and context from qualitative studies in two sections (a) Table 1 and (b) musculoskeletal complications (lines 126-128).

Table 1 illustrates a number of barriers - while this is useful - where is the data that supports this table?  Is this your opinion - or is it the synthesis of information from different studies?

RESPONSE: Thank you for the suggestion. We have mentioned it’s a summary in the text and now added references of the studies from which data was collected. (lines 219-220)

I think this manuscript has potential to be very useful - but it needs major revision as a review.

Reviewer 3 Report

Thank you very much for the opportunity to review this research. I think it is a very interesting proposal and that it should give continuity to the works published in pediatrics, but reflecting aspects of adulthood, and even the transition from adolescence to adulthood.

Therefore, I make the following suggestions.

Abstract

- No use of acronyms

1.Introduction

- Also use Peter Rosenbaum's reference on the definition of CP, since it is the most accepted

2. Causes of Cerebral Palsy

I leave a series of references that may be interesting

- Changing trends in cerebral palsy prevalence An opportunity to consider etiological pathways

- What constitutes cerebral palsy in the twenty-first century

3. Types of Cerebral Palsy

- Use functional classification systems, since they last until adulthood; and also to European classification and functional classification systems. In addition to the current classification based on topography (Validity and reliability of the guidelines of the surveillance of cerebral palsy in Europe for the classification of cerebral palsy.)

4. Aging with Cerebral Palsy

- Figure 1 can be completed with more things that are affected from pediatrics (Evidence-based diagnosis, health care, and rehabilitation for children with cerebral palsy)

4.1 The fact of not talking about hip dysplasia, where it is something almost mandatory, since it drags on from pediatrics and must be monitored

- The sections are very short of this point 4, it should be completed a little more

5. Benefits of Rehabilitation

-Table 1 is missing the abbreviations in its footer

- There are missing references in the text that appear in the references section. Effect of muscle strength training in children and adolescents with spastic cerebral palsy: A systematic review and meta-analysis

5.1 You should cite the works of Novak et al.; since if there are interventions that give functional changes

Regards

I think that English is good but minor editing of English language required

Author Response

REVIEWER #3

I make the following suggestions.

Abstract - No use of acronyms

RESPONSE: Thank you for the suggestion. We have removed all acronyms from the abstract.

1.Introduction- Also use Peter Rosenbaum's reference on the definition of CP, since it is the most accepted

RESPONSE: Thank you for the suggestion. We have updated the reference to cite P. Rosenbaums for the definition of CP (line 33)

  1. Causes of Cerebral Palsy

I leave a series of references that may be interesting

- Changing trends in cerebral palsy prevalence An opportunity to consider etiological pathways

- What constitutes cerebral palsy in the twenty-first century

RESPONSE: Thank you for the suggestion. “changing trends.....etiological pathways” was a commentary on an article by Smithers‐Sheedy. We have cited the original article “Declining trends in birth prevalence and severity of singletons with cerebral palsy of prenatal or perinatal origin in Australia: A population-based observational study” and “what constitutes cerebral palsy in the twenty first century” (lines 52-53).

  1. Types of Cerebral Palsy - Use functional classification systems, since they last until adulthood; and also to European classification and functional classification systems. In addition to the current classification based on topography (Validity and reliability of the guidelines of the surveillance of cerebral palsy in Europe for the classification of cerebral palsy.)

RESPONSE: Thank you for the suggestion. We have edited the section on types of CP to include the GMFCS and SCPE classification (lines 66-92)

  1. Aging with Cerebral Palsy - Figure 1 can be completed with more things that are affected from pediatrics (Evidence-based diagnosis, health care, and rehabilitation for children with cerebral palsy)

RESPONSE: We have added hip dysplasia, as this often begins in childhood. We feel that the other symptoms in the figure cover those that are affected from the time of pediatrics.

4.1 The fact of not talking about hip dysplasia, where it is something almost mandatory, since it drags on from pediatrics and must be monitored

- The sections are very short of this point 4, it should be completed a little more

RESPONSE: Thank you for the suggestion. We have added text on dysplasia (lines 126-130)

  1. Benefits of Rehabilitation

-Table 1 is missing the abbreviations in its footer

RESPONSE: Thank you for the suggestion. We have removed all abbreviations from the table.

- There are missing references in the text that appear in the references section. Effect of muscle strength training in children and adolescents with spastic cerebral palsy: A systematic review and meta-analysis

RESPONSE: Thank you for the suggestion. We have re-checked and updated our references.

5.1 You should cite the works of Novak et al.; since if there are interventions that give functional changes

RESPONSE: Thank you for the suggestion. We have cited the work of Novak et al for green light interventions (line 128-130 - dietary supplements and hip surveillance programs to improve bone mineral density and maintain hip integrity in adults).

Round 2

Reviewer 2 Report

Thank-you for completing the revisions to the paper - it is much improved with more comprehensive referencing.

My main concern continues to be with what type of paper this is. There is no methodology for the review - and it appears to fall into the narrative review category.  You refer to a review being completed in 2023 that is a systematic review and meta-analysis of chronic health conditions in adults with CP - and then you refer to your own paper as a review - when they are very different types of paper.

I do not know whether Brain Sciences considers this to be a review paper - but it would be more accurate to state clearly in the abstract that the intent of this manuscript is to highlight and discuss health challenges for adults with CP - and if it is to be considered a review, then it should be described as a narrative review.

As I pointed out in my previous review, no methodology has been identified for determining either the topics included, or the references selected. I think this paper is a worthwhile addition to the literature - but it should be made clear that it is neither a scoping or a systematic review.

I would recommend not using the word review so frequently in the manuscript, as many readers will have a different expectation of a review paper - and instead use words such as ‘discuss’, ‘describe’ or ‘highlight’.

Minor comments:

The description of GMFCS levels is useful - although in describing GMFCS V - it should be noted that some individuals at GMFCS V are able to achieve independent mobility in some environments using power mobility with extensive adaptations. In addition, GMFCS IV do not use body support walkers in most environments. They use wheeled mobility (and are independent with power mobility) in most environments, and can mobilize with body support walkers in some environments.

Author Response

Thank-you for completing the revisions to the paper - it is much improved with more comprehensive referencing.

My main concern continues to be with what type of paper this is. There is no methodology for the review - and it appears to fall into the narrative review category.  You refer to a review being completed in 2023 that is a systematic review and meta-analysis of chronic health conditions in adults with CP - and then you refer to your own paper as a review - when they are very different types of paper. I do not know whether Brain Sciences considers this to be a review paper - but it would be more accurate to state clearly in the abstract that the intent of this manuscript is to highlight and discuss health challenges for adults with CP - and if it is to be considered a review, then it should be described as a narrative review.

As I pointed out in my previous review, no methodology has been identified for determining either the topics included, or the references selected. I think this paper is a worthwhile addition to the literature - but it should be made clear that it is neither a scoping or a systematic review.

I would recommend not using the word review so frequently in the manuscript, as many readers will have a different expectation of a review paper - and instead use words such as ‘discuss’, ‘describe’ or ‘highlight’.

RESPONSE: Thank you for your helpful suggestion. We have edited the paper to specify that it is a narrative review. This has been added to the abstract and introduction. Where the prior draft said,”review,” the new draft says,”narrative review.” (lines 10, 13, 46). We also added text in the abstract to specify that our focus is on healthcare challenges (lines 10, 11). We also substituted in the word “discuss” in the text, line 46.

Minor comments:

The description of GMFCS levels is useful - although in describing GMFCS V - it should be noted that some individuals at GMFCS V are able to achieve independent mobility in some environments using power mobility with extensive adaptations. In addition, GMFCS IV do not use body support walkers in most environments. They use wheeled mobility (and are independent with power mobility) in most environments, and can mobilize with body support walkers in some environments.

RESPONSE: Thank you for your comments. We have revised GMFCS IV and V to read “GMFCS IV: the person typically uses wheeled mobility in most environments and may use body support walkers to mobilize in some environments. GMFCS V: the person is completely dependent for assistance in daily activities. Some might achieve self-mobility using a powered wheelchair with extensive adaptations”.